# Fertility-sparing surgery with neoadjuvant chemotherapy in early and locally advanced cervical cancer: A clinical protocol

**Momoko Tanioka, Shoji Nagao** ⓘ *, **Naoyuki Ida, Yui Tanaka, Atsushi Fujikawa, Ryoko Imatani, Yoshinori Tani, Hanako Sugihara, Kazuhiro Okamoto** ⓘ, **Hirofumi Matsuoka, Junko Haraga, Chikako Ogawa, Keiichiro Nakamura, Hisashi Masuyama**

Department of Obstetrics and Gynecology, Faculty of Medicine, Dentistry and Pharmaceutical Sciences, Okayama University, Okayama, Okayama, Japan

* s_nagao@okayama-u.ac.jp

## Abstract

Fertility preservation remains a critical concern in young women with early or locally advanced cervical cancer, as standard radical treatments compromise reproductive potential. This study aims to evaluate the feasibility, oncological safety, and reproductive outcomes of fertility-sparing treatment involving neoadjuvant chemotherapy followed by cervical conization and laparoscopic pelvic lymphadenectomy. This single-center, prospective, open-label, single-arm, Phase II interventional study will assess patients with FIGO stage IB2–IB3 cervical cancer (FIGO stage 2018) desiring fertility preservation. Eligible patients will receive three cycles of dose-dense paclitaxel and carboplatin (dd-TC), followed by conization and laparoscopic lymphadenectomy. The primary endpoint is successful uterine preservation. Patients requiring concurrent chemoradiotherapy due to inadequate treatment response will not be considered successful. Secondary endpoints include 2-year recurrence-free survival (RFS), overall survival (OS), quality of life assessments, menstrual and ovulatory resumption, pregnancy, live birth, miscarriage, and preterm birth. Adverse events will be graded according to CTCAE v5.0.

## Introduction

Cervical cancer remains a substantial global health burden, with an estimated 604,000 new cases and 342,000 deaths reported worldwide in 2020 [1]. In Japan, approximately 10,000 new cases and 3,000 deaths occur annually, with 42% of cases affecting women under 45 years of age [2]. Given the disproportionate impact on women of reproductive age, fertility preservation has become increasingly critical in early-stage cervical cancer management.

**Data availability statement:** No datasets were generated or analysed during the current study. All relevant data from this study will be made available upon study completion.

**Funding:** This study is supported by an Okayama University Hospital Clinical Research Incentive Grant.

**Competing interests:** The authors declare that they have no competing interests.

For patients with FIGO stage IB1disease (FIGO stage 2018), recent evidence has shifted the standard of care from radical hysterectomy to a less radical approach. The SHAPE trial demonstrated that simple hysterectomy combined with pelvic lymph node assessment was noninferior to radical hysterectomy in terms of 3-year pelvic recurrence rates (2.52% vs. 2.17%) and was associated with significantly reduced surgical morbidity [3]. The ConCerv trial prospectively evaluated conservative surgeries such as conization with lymph node assessment in patients with low-risk early-stage disease, confirming oncologic safety, with a recurrence rate of 3.5% and a lymph node metastasis rate of 5% [4]. Most recently, the Gynecologic Oncology Group (GOG) in United States conducted the GOG-278 trial, which examined the oncological adequacy, contribution to physical function and quality of life, and reproductive outcomes of simple hysterectomy or cone biopsy plus pelvic lymphadenectomy for early stage IA1-IB1 cervical cancer [5,6]. After a median follow-up of 37 months in 201 eligible patients, only 3 (4%) cone biopsy patients have experienced disease recurrence. In addition, patients treated with cone biopsy plus pelvic lymphadenectomy achieved successful pregnancies. Incorporating these findings from SHAPE, ConCerv, and GOG-278 studies, less invasive surgical options are reasonable for appropriately selected patients with tumors ≤2 cm [3–6].

However, for patients with tumors 2–4 cm in size, radical hysterectomy remains the standard treatment, and the safety of fertility preservation strategies has not been fully established. Although radical trachelectomy with pelvic lymph node dissection can preserve fertility while ensuring oncological safety. However, the postoperative pregnancy rate is lower than that of conization, and the procedure is associated with higher rate of miscarriage and preterm birth [7].

Furthermore, for locally advanced cervical cancer, including patients with tumors >4 cm, neoadjuvant chemotherapy (NAC) followed by surgery has been investigated as a potential strategy for less extensive surgery. Large, randomized trials, including the TATA study and EORTC 55994, have not demonstrated a survival advantage of NAC followed by radical hysterectomy over concurrent chemoradiation therapy (CRT); consequently, NAC is not currently recommended as the standard treatment for locally advanced cervical cancer [8,9]. Although these studies focused on the efficacy of NAC in terms of survival compared to CRT, the potential use of NAC as a downstaging strategy to facilitate fertility-sparing surgery remains unexplored.

Cervical cancer is highly chemosensitive and dose-dense chemotherapy regimens can achieve high tumor response rates. Our group previously conducted a phase II trial to investigate a multidisciplinary strategy utilizing dose-dense paclitaxel and carboplatin (dd-TC) administered both before and after radical hysterectomy in patients with locally advanced cervical cancer [10]. This trial demonstrated an overall response rate of 92%, including a 22% pathological complete response rate. Notably, 98% of the patients successfully underwent planned radical hysterectomy without requiring postoperative radiotherapy.

Based on these promising results, we hypothesize that dd-TC therapy can effectively downstage tumors >2 cm to ≤2 cm, potentially enabling oncologically safe conization as a fertility-sparing procedure for patients who would otherwise be ineligible.

Recently, several case series have reported the oncological and obstetric outcomes of NAC followed by cervical conization strategy for FIGO stage IB2-IB3 (FIGO stage 2018) cervical cancer, but no prospective studies have been reported to date [11,12]. This innovative approach may expand fertility preservation options, reduce the reliance on radical trachelectomy, and improve reproductive and obstetric outcomes.

## Patients and methods (Figs 1 and 2)

### 1. Study design

This is a single-center, prospective, open-label, single-arm, Phase II interventional study that will prospectively enroll eligible participants. All eligible participants will undergo the specified interventions.

### 2. Participants

**Inclusion criteria.** Participants are eligible if they meet all of the following:

- Clinical diagnosis of cervical cancer, FIGO stage IB2 or IB3 according to the 2018 classification

- Histopathological confirmation of squamous cell carcinoma, adenocarcinoma, or adenosquamous carcinoma of the cervix

- Pre-menopausal status, confirmed by menstrual history and/or hormonal assays if ambiguous

- Age ≤ 40 years at enrollment (to ensure high likelihood of ovarian function and successful pregnancy outcomes)

- Adequate systemic organ function suitable for chemotherapy administration

| | STUDY PERIOD | | | | | | | |
| --- | --- | --- | --- | --- | --- | --- | --- | --- |
| | Enrolment | | INTEVENTION | | | | | FOLLOW-UP** |
| TIME POINT (weeks) | -4 to -2 | 0 | 0 to 3 | 3 to 6 | 6 to 9 | 11 to 13 | 15 to 17 | |
| ENROLMENT: | | | | | | | | |
| Eligibility screen | × | | | | | | | |
| Informed consent | × | | | | | | | |
| Registration | | × | | | | | | |
| INTERVENTIONS: | | | | | | | | |
| dose dense TC therapy | | | × | × | × | | | |
| Cervical conization | | | | | | × | | |
| laparoscopic pelvic lymphadenectomy | | | | | | | × | |
| ASSESSMENTS: | | | | | | | | |
| Physical findings | × | | × | × | × | | | × |
| Pelvic MRI | × | | × | × | × | | | |
| FDG-PET | ×* | | | | | | | |
| CT scan | | | | | × | | | × |
| Histopathological examination | × | | | | | × | × | |
| Cervical cytology | × | | | | | | | × |
| QOL survey*** | × | × | × | | | | × | × |

*CT scan is acceptable as a substitute.

**Conducted every 3 months for 2 years.

***QOL survey includes Functional Assessment of Cancer Therapy–Cervix, Female Sexual Function Index, and Hospital Anxiety and Depression Scale.

**Fig 1. SPIRIT schedule of Fertility-preserving treatment.**

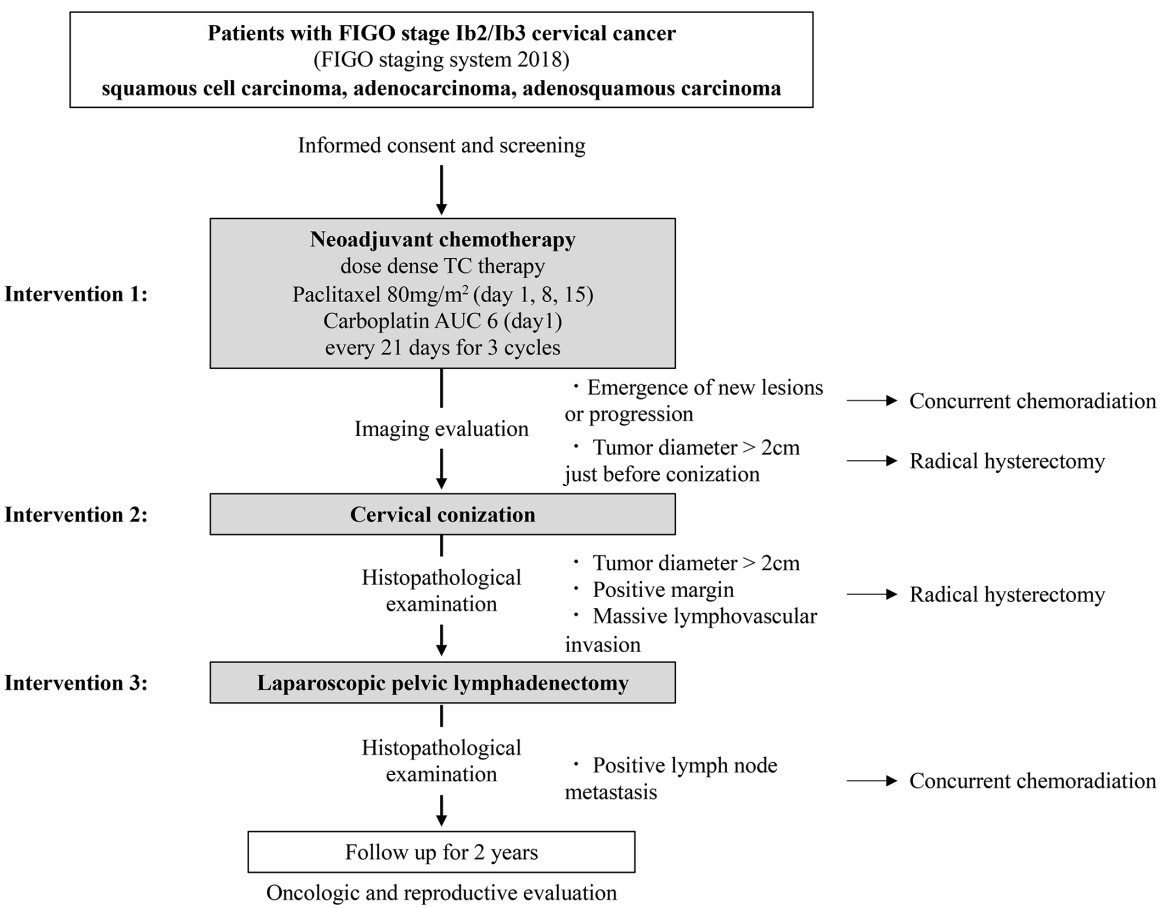

**Fig 2. Fertility-preserving treatment algorithm.** AUC: area under the curve.

- Strong desire for future fertility preservation, explicitly expressed by the patient

- Written informed consent after thorough explanation of study procedures, potential risks, and benefits

  **Exclusion criteria.** Participants are excluded if any of the following apply:

- Age > 40 years

- HPV-independent cervical cancer

- Active synchronous malignancy requiring treatment

- Severe comorbidities that contraindicate chemotherapy or surgery, or significantly compromise ability to complete the study protocol

- Documented history of hypersensitivity reactions to formulations containing polyoxyethylene castor (Cremophor EL) or hydrogenated castor oil

- Active infections requiring systemic antimicrobial therapy

- Pregnancy, lactation, or possibility of pregnancy

- Any other condition that could render the patient unsuitable for study participation

Imaging assessments will generally include pelvic contrast-enhanced magnetic resonance imaging (MRI) and 18F-fluorodeoxyglucose positron emission tomography (FDG-PET). Tumor size and presence or absence of lymph node metastasis will be independently evaluated by one imaging specialist and one gynecologic oncologist. Lymph node metastasis will be considered suspected if the maximum standardized uptake value (SUVmax) exceeds 2.5 or if the short-axis diameter of the lymph node is greater than 10 mm. In cases of discordance between the two reviewers, the final diagnosis will be determined through consensus discussion.

### 3. Interventions

The study protocol will encompass the following three phases.

**Intervention 1: Neoadjuvant chemotherapy (three cycles of dd-TC).** Participants will receive paclitaxel 80 mg/m² intravenously on days 1, 8, and 15, combined with carboplatin at an area under the curve (AUC) of 6 administered intravenously on day 1. Each cycle will be repeated every 21 days [10]. The paclitaxel dosage is calculated based on body surface area using the DuBois formula, whereas the carboplatin dose is determined using the Calvert formula, considering renal function [13,14].

Detailed guidelines will be established for chemotherapy initiation, dose-reduction procedures for toxicity, management of paclitaxel-related hypersensitivity reactions (including premedication strategies), criteria for recalculating drug dosages, and termination or temporary suspension of treatment based on toxicity or disease progression.

Supportive care will be permitted within predefined limits, including granulocyte colony-stimulating factor (G-CSF) for myelosuppression, iron supplements, blood transfusions as needed, antiemetics for nausea and vomiting, and anti-allergic agents. Concomitant anticancer treatments, including radiotherapy, other chemotherapy regimens, immunotherapy, hormonal therapy, and investigational drugs, are strictly prohibited.

A pelvic contrast-enhanced MRI will be performed after each chemotherapy cycle. Prior to conization, a contrast-enhanced computed tomography (CT) scan and, if clinically indicated, an 18F- FDG-PET scan will also be conducted. Changes in tumor size and the presence or absence of distant metastases, including lymph node involvement, will be independently assessed by one diagnostic radiologist and one gynecologic oncologist. In cases of disagreement, the final assessment will be determined through consensus discussion.

**Intervention 2: Cervical conization.** After completion of NAC, eligible patients will undergo cervical conization. Eligibility will require reassessment of tumor response, including pelvic MRI and, if necessary, PET-CT, confirming regression to ≤2 cm with no new lesions. MRI must demonstrate an intact stromal margin of at least 3 mm and a stromal invasion depth <50%. The decision will be made separately by one imaging specialist and one gynecological oncologist, and if there is no agreement, the final decision will be made by discussion.

After ligating the descending branch of the uterine artery, the tumor will be excised with a margin of at least 5 mm. In patients with no visible tumor (e.g., complete response to NAC), a sufficient portion of the cervix, typically 2 cm in depth, will be excised to ensure adequate surgical margins. If postoperative pathology reveals a maximum residual tumor diameter ≤2 cm with a positive or indeterminate margin, repeat cervical conization will be considered. When conization is performed twice, the cumulative tumor diameter and stromal invasion depth will be used for risk assessment.

If the pathological tumor size is ≤2 cm, with no massive lymphovascular invasion and negative margins for invasive cancer, the patient will proceed to the subsequent intervention. If the tumor size remains >2 cm, stromal invasion is >10 mm, or margins are positive, open radical hysterectomy will be performed as definitive treatment.

**Intervention 3: Laparoscopic pelvic lymphadenectomy.** Patients who meet the pathological criteria after conization (tumor size ≤2 cm, stromal invasion <10 mm, and negative margins) will undergo laparoscopic pelvic lymphadenectomy for accurate staging and treatment decisions.

During the laparoscopic procedure, if disseminated disease, ovarian metastasis, or grossly enlarged lymph nodes are identified, or if rapid pathology reveals nodal metastasis, the fertility-sparing pathway will be discontinued. Subsequent treatment will be determined according to routine clinical practice. If definitive pathology confirms lymph node metastasis, fertility preservation will be abandoned, and the patient will receive CRT with weekly cisplatin.

## 4. Quality of life and toxicity assessment

Patient-reported outcomes will be assessed using validated questionnaires: Functional Assessment of Cancer Therapy–Cervix (FACT-Cx), Female Sexual Function Index (FSFI), and Hospital Anxiety and Depression Scale (HADS). Assessments will be conducted at baseline and at prespecified intervals.

All adverse events will be systematically collected, graded according to CTCAE v5.0, and continuously reported. Laboratory tests (CBC; liver and renal function) will be performed regularly to monitor toxicities.

## 5. Follow-up and outcome assessment

Post-treatment surveillance will be standardized to evaluate oncologic control and functional outcomes. During the first 2 years, participants will undergo cervical cytology every 3 months and contrast-enhanced CT every 6 months; thereafter, cytology will be performed every 6 months and CT annually for 3 years. Recurrence is defined as the date on which invasive carcinoma is confirmed clinically or pathologically; CIN alone will not be considered recurrence. Female health-related QOL will be assessed using FACT-Cx, FSFI, and HADS. Menstrual and ovulatory status will be documented from histories and basal body temperature charts.

## 6. Sample size

A target sample size of 10 patients is set for this single-arm Phase II study. This sample will provide preliminary data on efficacy and safety, allowing consideration of a future Phase III trial. An interim analysis will be conducted after 5 patients have been enrolled and their primary outcomes assessed. If uterine preservation is ≤ 2 cases, enrollment will be suspended and feasibility will be re-evaluated.

## 7. Statistical analysis

The full analysis set will include all participants who initiate study treatment with at least one post-baseline assessment. The safety population will include all participants who receive at least one study dose. Data will be entered into a secure, non-networked system. Descriptive statistics will be calculated for demographics, baseline characteristics, and study endpoints.

The primary endpoint, uterine preservation rate, will be calculated with a corresponding confidence interval. Secondary endpoints, including RFS and OS, will be estimated using Kaplan–Meier methods. Menstrual resumption, ovulation, pregnancy, live birth, miscarriage, and preterm birth will be reported as proportions. Toxicities will be summarized by type, grade, and frequency.

Missing data for patient-reported outcomes will be handled using the last observation carried forward method. For survival endpoints and pregnancy-related outcomes, data collection will continue as comprehensively as possible throughout the study period. Because this is an exploratory study, a formal sample size calculation will not be conducted. The target sample size of 10 was chosen based on practical considerations regarding the rarity of FIGO stage IB2–IB3 patients seeking fertility preservation at a single center, as well as the ethical necessity of limiting participant exposure to this novel approach until the planned interim safety analysis is completed. If the number of events allows, exploratory multivariate analyses will be performed using Cox proportional hazards regression.

## 8. Timeline

The study will commence upon approval by the Certified Clinical Research Review Board and is planned to conclude on March 31, 2030.

## 9. Ethical considerations

The research will adhere to the Declaration of Helsinki, the Ethical Guidelines for Medical Research Involving Human Subjects in Japan, and other applicable national and international regulations. This study was approved by the Certified Clinical Research Review Board of Okayama University (approval number: CRB25−005). Written informed consent will be obtained from all participants prior to their participation in this study. During the informed consent process, it is explicitly emphasized to all participants that this treatment strategy is non-standard and exploratory in nature, involving potential risks associated with the dual-experimental approach.

## 10. Study registration

This study was registered with the Japan Registry of Clinical Trials and published on July 24, 2025 (clinical research protocol number: jRCTs061250041).

## Results and discussion

This study aimed to evaluate the feasibility, oncological safety, and reproductive outcomes of a novel fertility-preserving treatment strategy for women with early and locally advanced cervical cancer without parametrial invasion (FIGO 2018 stage IB2–IB3) who desired to retain their reproductive potential. We hypothesized that neoadjuvant dd-TC could effectively downstage tumors larger than 2 cm to ≤2 cm, thereby facilitating oncologically safe cervical conization with laparoscopic pelvic lymphadenectomy. Notably, the target patients are different from those in the SHAPE, ConCerv, and GOG-278 trials, which initially targeted patients with tumors ≤2 cm [3–6]. This approach is designed to expand fertility preservation options for patients who would otherwise require radical hysterectomy or definitive chemoradiotherapy, both of which result in permanent infertility.

This study incorporates two experimental approaches: NAC using dd-TC and less radical surgery aimed for fertility preservation. While this presents a dual experimental approach, the favorable oncological outcomes of less radical surgery in stage IA–IB1 cervical cancer have already been demonstrated in multiple clinical studies [3–6]. Therefore, the risk of combining less radical surgery with dd-TC, which has a proven high response rate, is considered acceptable provided that strict discontinuation criteria are established. Our protocol incorporates radiologic and histopathologic assessments at multiple time points to ensure patient safety. The primary objective of this study is to determine whether cervical cancers downstaged to <2 cm by NAC can be treated identically to cervical cancers originally <2 cm in size. Importantly, successful uterine preservation in this study is not viewed as an isolated metric but is evaluated in close conjunction with oncological safety endpoints, such as 2-year RFS, to ensure that fertility-sparing does not jeopardize patient survival.

In this protocol, instead of sentinel lymph node biopsy (SLNB) and ultrastaging, we performed radiological lymph node evaluation using PET-CT and MRI after NAC was performed, followed by pathological confirmation via laparoscopic complete pelvic lymphadenectomy after cervical conization. However, we recognize the inherent limitations of imaging-based assessment, as PET-CT and MRI may not fully detect occult micro-residual disease or small-volume nodal involvement after NAC. The rationale for this is that, at the time of the study initiation, only Techne® Phytate (used in the RI method) was covered by public health insurance in Japan, and the SLNB procedure itself had not yet been approved. Although the fluorescent dye tracer indocyanine green was subsequently approved in July 2025, and SNLB approval is expected to follow, there are currently no data regarding the efficacy and safety of performing SLNB either before or after dd-TC.

Introducing SLNB would therefore constitute an additional experimental variable. Consequently, the future application of SLNB in this study requires careful discussion.

Patients with FIGO stage IB2–IB3 cervical cancer (FIGO stage 2018), inherently carry an estimated recurrence risk of approximately 10% [15,16]. Accordingly, we prioritized patient safety throughout both the interventions and study procedures. To this end, explicit criteria were established to determine eligibility for proceeding to each subsequent intervention step. Additionally, an interim safety analysis was planned after the enrollment of the first five patients. Depending on the results of this interim evaluation, the study design allowed for protocol modification or, if deemed necessary, trial discontinuation to safeguard participants [17].

The small sample size, single-center design, and relatively short follow-up period (2 years) inherently limit the generalizability of this feasibility study. However, these limitations are acceptable given that the present study was designed as an exploratory investigation to generate preliminary data for future multicenter phase II or III trials. We believe that the present design represents a practical and feasible framework for evaluating the safety and preliminary efficacy of this novel fertility-sparing approach.

This study aims to clarify the oncological safety and obstetric outcomes of this novel fertility-preserving approach, and to identify the types of assisted reproductive technology required. Medically, this study may establish a new treatment paradigm that balances oncologic control with fertility preservation in patients currently deemed ineligible for conservative surgery, thereby improving the quality of life and long-term reproductive outcomes. Furthermore, we intend to assess its social acceptability and determine the extent of patient and societal demand for such fertility-preserving strategies. From a societal perspective, this study may reduce the psychological and economic burdens associated with infertility in young women with cervical cancer, facilitate family planning, and contribute to broader reproductive health equity. These findings may also inform international guidelines and stimulate further research into fertility-sparing strategies for bulky cervical cancer.

## Supporting information

**S1 File. SPIRIT 2025 Checklist.**
(DOCX)

**S2 File. FepCC study protocol version 3.0 (original).**
(DOCX)

**S3 File. FepCC study protocol version 3.0 (English).**
(DOCX)

**S4 File. FepCC study explanation and consent form version 3.0 (original).**
(DOCX)

**S5 File. FepCC study explanation and consent form version 3.0 (English).**
(DOCX)

## Acknowledgments

We thank Editage (www.editage.jp) for their assistance with English language editing.

## Author contributions

**Conceptualization:** Momoko Tanioka, Shoji Nagao, Naoyuki Ida.

**Funding acquisition:** Momoko Tanioka, Naoyuki Ida.

**Supervision:** Hisashi Masuyama.

**Writing – original draft:** Shoji Nagao.

**Writing – review & editing:** Momoko Tanioka, Shoji Nagao, Naoyuki Ida, Yui Tanaka, Atsushi Fujikawa, Ryoko Imatani, Yoshinori Tani, Hanako Sugihara, Kazuhiro Okamoto, Hirofumi Matsuoka, Junko Haraga, Chikako Ogawa, Keiichiro Nakamura, Hisashi Masuyama.

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
