## [Decision Letter · Decision Letter 0]

31 Oct 2025

Dear Dr. Nagao,

Thank you for submitting your manuscript to PLOS ONE. After careful consideration, we feel that it has merit but does not fully meet PLOS ONE’s publication criteria as it currently stands. Therefore, we invite you to submit a revised version of the manuscript that addresses the points raised during the review process.

**Major Comments:**

**Relation to recent GOG studies:**

Two recent GOG phase II trials (Gynecol Oncol. 2025;195:50–58, 59–65) have explored fertility-sparing strategies combining conization and lymphadenectomy. These should be cited and discussed to contextualize the study design and highlight differences or innovations in your protocol.

**Dual experimental approach:**

The concurrent use of NAC and fertility-sparing surgery represents a “double experimental” design. The rationale for combining these interventions should be explicitly justified, and the potential risks of this approach discussed.

**Oncologic staging:**

The protocol does not include a **pre-treatment sentinel lymph node biopsy (SLNB)** , which is critical for accurate baseline staging. Please justify this omission or consider including SLNB (with ultrastaging) as a criterion for eligibility (pN0, macro- or micrometastasis).

**Endpoints and limitations:**

The primary endpoint (uterine preservation) is clinically meaningful but should be interpreted within the constraints of a short 2-year follow-up, single-center design, and very small sample size (n=10). These limitations should be emphasized in the Discussion.

**Statistical plan:**

Include more detail regarding missing data handling, adjustment for confounders, and justification for the sample size (even if feasibility-based).

**Novelty and contribution:**

The use of **dose-dense paclitaxel/carboplatin (dd-TC)** NAC prior to fertility-sparing surgery could represent a distinctive contribution. Please emphasize this and clearly state how this approach differs from previous reports.

**Clinical implications:**

The Discussion should better outline how this feasibility study can inform future multicenter trials and guideline development.

**Minor Comments:**

Clarify imaging modalities for determining lymph node status (PET-CT vs. CT).

Specify how tumor size is measured and reviewed (MRI criteria, double reading).

Correct minor grammatical inconsistencies.

Add line numbering for easier reference during revision.

We look forward to receiving your revised manuscript.

Kind regards,

Kazunori Nagasaka

Academic Editor

PLOS ONE

Journal Requirements:

2. Figures 1 and 2 in the "study_protocol_E.docx" SI file were provided in non-English language. Please provide an updated “study_protocol_E.docx” with Figures 1 and 2 in English language.

“Okayama University Hospital Clinical Research Incentive Grant”

**Additional Editor Comments:**

Dear Authors,

This manuscript presents a prospective, single-arm phase II trial evaluating fertility-sparing treatment with neoadjuvant dose-dense paclitaxel and carboplatin followed by conization and laparoscopic pelvic lymphadenectomy for FIGO stage IB2–IB3 cervical cancer. The study is well organized and clinically meaningful, addressing a highly relevant but challenging area in gynecologic oncology.

However, several important revisions are required before publication. The protocol should be better contextualized with recent literature, particularly the two GOG phase II studies published in Gynecologic Oncology (2025;195:50–58, 59–65), which explored similar fertility-sparing strategies. A comparative discussion is needed to clarify the novelty of the present trial.

The study combines two experimental elements—neoadjuvant chemotherapy and conservative surgery—and the rationale for this dual approach should be explained more clearly, including safety considerations. The lack of sentinel lymph node biopsy before treatment is a notable limitation that may affect staging accuracy; justification or modification should be discussed.

While the primary endpoint (uterine preservation) is clinically appropriate, the small sample size (n=10), single-center design, and short follow-up substantially limit generalizability. The statistical plan requires more detail, including handling of missing data, adjustment for confounders, and justification for sample size, even if based on feasibility.

The Discussion should emphasize the distinctiveness of the dose-dense paclitaxel/carboplatin regimen and elaborate on how this feasibility study could inform future multicenter or randomized trials.

Minor grammatical corrections and line numbering are also recommended.

Overall, this is a promising and well-structured clinical protocol that addresses an important gap in fertility preservation for cervical cancer.

With the above revisions, it would make a stronger and more rigorous contribution.

Sincerely,

Plos One Editorial Office

Reviewers' comments:

Reviewer's Responses to Questions

**Comments to the Author**

1. Does the manuscript provide a valid rationale for the proposed study, with clearly identified and justified research questions?

Reviewer #1: Yes

Reviewer #2: Yes

Reviewer #3: Yes

Reviewer #4: Partly

Reviewer #5: Yes

2. Is the protocol technically sound and planned in a manner that will lead to a meaningful outcome and allow testing the stated hypotheses?

Reviewer #1: Partly

Reviewer #2: No

Reviewer #3: Partly

Reviewer #4: No

Reviewer #5: Yes

3. Is the methodology feasible and described in sufficient detail to allow the work to be replicable?

Reviewer #1: Yes

Reviewer #2: No

Reviewer #3: Yes

Reviewer #4: No

Reviewer #5: Yes

4. Have the authors described where all data underlying the findings will be made available when the study is complete?

Reviewer #1: Yes

Reviewer #2: No

Reviewer #3: Yes

Reviewer #4: No

Reviewer #5: Yes

5. Is the manuscript presented in an intelligible fashion and written in standard English?

Reviewer #1: Yes

Reviewer #2: Yes

Reviewer #3: Yes

Reviewer #4: Yes

Reviewer #5: Yes

You may also provide optional suggestions and comments to authors that they might find helpful in planning their study.

Reviewer #1: This manuscript presents a prospective, single-arm phase II trial investigating fertility-sparing treatment with neoadjuvant chemotherapy (NAC) followed by conization and laparoscopic pelvic lymphadenectomy for women with FIGO stage IB2–IB3 cervical cancer. The focus on oncologic safety alongside reproductive outcomes is clinically meaningful, and the detailed description of endpoints is commendable.

Nevertheless, several concerns should be addressed to strengthen the manuscript:

1. Relation to recent GOG studies: Two recent GOG trials (Gynecol Oncol. 2025;195:50-58, 59-65) investigated fertility-sparing strategies using conization and laparoscopic lymphadenectomy. These important studies are not cited, and their omission limits the contextualization of the current work. The authors should clearly compare their design with the GOG protocols, highlighting similarities and differences.

2. Dual experimental approach: The trial combines two experimental elements: (i) NAC as induction therapy and (ii) fertility-sparing surgery (conization plus lymphadenectomy). While each strategy has some supporting evidence, their concurrent use introduces complexity and potential risk. It will be important for the authors to acknowledge this “double experimental arm” and to discuss whether observed outcomes can be attributed to NAC, the surgical approach, or their combination. A clear rationale and safety considerations for using both interventions simultaneously should be explicitly stated.

3. Endpoints and limitations: The primary endpoint of successful uterine preservation is clinically relevant, but its adequacy must be interpreted in the context of long-term oncologic outcomes. The relatively short follow-up period (2-year RFS) and the single-center nature of the trial are significant limitations that need to be clearly recognized.

4. Novelty and contribution: The incorporation of dose-dense paclitaxel/carboplatin NAC before fertility-sparing surgery could represent a unique contribution compared with prior trials, but the authors should emphasize this distinction more strongly.

5. Clinical implications: The discussion should elaborate on how this trial may inform future multicenter investigations or guideline development, particularly if preliminary safety and feasibility are confirmed.

In summary, this is a timely and clinically relevant protocol. However, the manuscript should integrate recent GOG findings, acknowledge the dual experimental nature of the intervention, and clarify the trial’s novelty, limitations, and future implications.

Reviewer #2: This clinical protocol describes a Phase II clinical trial (n=10) to evaluate a fertility-preserving treatment for women aged ≤40 years with FIGO stage IB2–IB3 cervical cancer. The primary objective is uterine preservation, with secondary outcomes including recurrence-free survival, overall survival, reproductive outcomes, and quality of life. The study investigates whether neoadjuvant dose-dense chemotherapy can downstage tumors larger than 2 cm, enabling safe conservative surgery in patients who would otherwise require radical treatment.

Minor Revisions Suggested:

1. Include a more detailed statistical analysis plan, addressing missing data handling, confounding variables, and sensitivity analyses.

2. Provide justification for the sample size, even if based on feasibility rather than formal hypothesis testing.

3. Add line numbering to the manuscript to facilitate peer review.

Limitations and Areas for Improvement:

1. Small Sample Size: The planned enrollment of 10 patients is suitable for a pilot feasibility study but limits statistical power and generalizability. No formal power calculation is provided to justify this sample size. Additionally, the small sample size is insufficient to reliably estimate outcomes related to the secondary objectives.

2. Lack of Control Group: As a single-arm study, the absence of a comparator group restricts the ability to assess relative efficacy or draw causal conclusions.

3. Handling of Missing Data: The protocol does not specify strategies for managing missing data, which is particularly important for longitudinal outcomes such as quality of life and reproductive metrics.

4. No Adjustment for Confounders: There is no mention of multivariate analysis or adjustment for potential confounding factors (e.g., tumor histology, age, baseline fertility status), which may affect outcome interpretation.

5. Limited Statistical Detail: The statistical analysis section lacks detail regarding the software to be used, assumptions underlying Kaplan–Meier analysis, and plans for subgroup or exploratory analyses.

Reviewer #3: This study addresses an important clinical challenge—balancing oncologic control and fertility preservation in younger women with cervical cancer. However, its single-arm, single-center design with a very small sample size significantly limits the strength of conclusions that can be drawn. The lack of a control arm prevents comparative effectiveness assessment. The stringent inclusion criteria and complex intervention pathway may also limit applicability.

The lack of routine pre-treatment sentinel lymph node biopsy is a major limitation of the current study protocol. It compromises accurate baseline staging, risks undertreatment, and may lead to unnecessary procedures with associated morbidity and psychological distress. Incorporating SLNB before initiating chemotherapy and surgery would enhance patient selection, improve oncologic safety, and optimize fertility preservation outcomes. Addressing this gap is essential for ensuring the study’s clinical relevance and ethical soundness.

To enhance robustness, the study could benefit from:

- Include laparoscopic pelvic Sentinel-Node Biopsy (with Ultrastaging) as a pre-treatment criteria

- Study Inclusion Criteria: pN0 (macro and micrometastasis)

- Including a matched control group or historical controls for comparison.

- Expanding sample size and considering multicenter involvement.

- More direct fertility-related endpoints (pregnancy/live birth rates).

- Clearer criteria for intervention discontinuation and handling non-responders.

- More detailed patient support to manage psychological impact.

Despite these limitations, as an early Phase II feasibility study, it is a reasonable first step toward defining fertility-preserving strategies in cervical cancer, provided its findings are interpreted cautiously.

Reviewer #4: Dear authors, I congratulate you on your idea, but it needs to be better contextualised in order to start the study protocol.

The main problem is related to tumour staging.

1- You need to clarify how you determine lymph node negativity (with PET? With lymphadenectomy? At what times and in what manner?)

2- You must specify how you will determine tumour size (double blind MRI review?).

3- You must specify whether you will exclude patients who have undergone conisation as a diagnostic procedure.

4- You must specify how you will deal with negative prognostic histological factors (invasion of lymphovascular spaces, deep stromal infiltration, g3, etc.).

5- The sample size presented is insufficient. Please set yourself an outcome to observe and build a sample size based on your objectives.

5- The conceptual part is outdated and should be updated with the latest knowledge on the subject (PMID: 37261562; PMID: 36064991).

Only after these necessary corrections have been made can the study be considered for publication.

Reviewer #5: Dear Author,

I have reviewed your article entitled: “Fertility-sparing surgery with neoadjuvant chemotherapy in early and locally advanced cervical cancer: A clinical protocol”. This manuscript describes a well-structured Phase II, single-arm, single-center clinical protocol investigating the feasibility and oncologic safety of fertility-sparing surgery following neoadjuvant dose-dense paclitaxel and carboplatin (dd-TC) in patients with FIGO 2018 stage IB2–IB3 cervical cancer. The topic is clinically relevant and of potential high impact, as fertility preservation in cervical cancer remains an unmet need for women with tumors larger than 2 cm.

I have some consideration:

• Do you perform PET-CT or CT scans for all patients? Could you clarify it?

• Extending or clarifying the reproductive follow-up period would improve the study’s relevance

• Minor grammatical consistency is recommended.

• I suggest to add some references like these: PMID: 39531915; PMID: 38471373

**Do you want your identity to be public for this peer review?** For information about this choice, including consent withdrawal, please see our Privacy Policy

Reviewer #1: **Yes: ** Tsukasa Baba

Reviewer #2: No

Reviewer #3: No

Reviewer #4: **Yes: ** Carlo Ronsini

Reviewer #5: No

---

## [Author Response · Author response to Decision Letter 1]

15 Nov 2025

We sincerely thank the Academic Editor and all reviewers for their thoughtful and constructive comments on our manuscript entitled “Fertility-sparing surgery with neoadjuvant chemotherapy in early and locally advanced cervical cancer: A clinical protocol.” We have carefully revised the manuscript in accordance with the feedback provided. A detailed, point-by-point response is presented below. All suggested changes have been incorporated into the revised version, and grammatical and formatting corrections have been made throughout.

Major Comments:

Relation to recent GOG studies:

Two recent GOG phase II trials (Gynecol Oncol. 2025;195:50–58, 59–65) have explored fertility-sparing strategies combining conization and lymphadenectomy. These should be cited and discussed to contextualize the study design and highlight differences or innovations in your protocol.

Response: Thank you for your very helpful comments. The GOG278 trial examined the oncological adequacy, contribution to physical function and quality of life, and perinatal prognosis of simple hysterectomy or cone biopsy plus pelvic lymphadenectomy for early stage IA1-IB1 cervical cancer. It was shown that cone biopsy plus pelvic lymphadenectomy can achieve a reasonable oncological prognosis while ensuring a favorable perinatal prognosis. Our study focused on patients with stage IB2 or IB3 disease whose tumor diameter was reduced to ≤2 cm following dose-dense TC therapy, representing a different patient population. Although the clinical implications of achieving a tumor diameter ≤2 cm through chemotherapy differ from those of patients with an initial tumor diameter ≤2 cm, we consider the GOG-278 data to provide supporting evidence for the validity of our approach.

We added some discussion in "Introduction" (line 49 to 57) and "Results and Discussion" section (line 238 to 241).

Dual experimental approach:

The concurrent use of NAC and fertility-sparing surgery represents a “double experimental” design. The rationale for combining these interventions should be explicitly justified, and the potential risks of this approach discussed.

Response: As you pointed out, we acknowledge the risks associated with the double experimental approach. Based on the results of SHAPE, ConCerv, and GOG-278, we believe that the oncological safety of less radical surgery for stage IA-IB1 cervical cancer, particularly conization, as investigated in the ConCerv and GOG-278 trials, has been confirmed to be oncologically safe to a certain extent, although the evidence remains limited when conization alone is considered. Furthermore, the SGSG016 study confirmed a high response rate of over 90% and a low PD rate for dose-dense TC chemotherapy, confirming that it is a suitable regimen for preoperative tumor reduction. As described above, we believe that the risks associated with the double experimental approach are within an acceptable range as long as measures are taken to adequately ensure participant safety. This study clearly defines methods for ensuring patient safety based on the results of imaging tests during dose-dense TC therapy and before conization, the histopathological results of the conization specimens, and the histopathological results of the pelvic lymph node dissection specimens. Furthermore, to address the potential risks to ovarian function caused by chemotherapy, it is recommended that embryo freezing or oocyte freezing be performed before starting dose-dense TC therapy.

We have added some discussion in "Results and Discussion" section (line 263 to 277).

Oncologic staging:

The protocol does not include a pre-treatment sentinel lymph node biopsy (SLNB), which is critical for accurate baseline staging. Please justify this omission or consider including SLNB (with ultrastaging) as a criterion for eligibility (pN0, macro- or micrometastasis).

Response: Although sentinel lymph node biopsy (SLNB) with ultrastaging provides accurate nodal assessment, nodal status will be assessed radiologically (PET-CT and MRI) before NAC and confirmed pathologically by laparoscopic complete pelvic lymph node dissection after conization in this protocol. It was not included in this protocol for three reasons: first, it avoids additional invasive procedures before chemotherapy and maintains the feasibility of treatment. Second, SLNB after NAC is not currently standard practice and further increases the experimental approach. Finally, SLNB for early-stage cervical cancer is not currently covered by public insurance in Japan. If SLNB becomes eligible for insurance coverage in Japan in the future, it may be possible to consider SLNB for some eligible patients.

We added the discussion regarding SLNB in "Results and Discussion" section (line 278 to 286).

Endpoints and limitations:

The primary endpoint (uterine preservation) is clinically meaningful but should be interpreted within the constraints of a short 2-year follow-up, single-center design, and very small sample size (n=10). These limitations should be emphasized in the Discussion.

Response: As you pointed out, the small sample size, single-center design, and short follow-up period (2 years) inherently limit the generalizability of this feasibility study. However, these limitations are appropriate as an exploratory study aimed at generating pilot data for future multicenter Phase II/III trials. There are uncertainties, such as the extent to which this approach will be accepted by Japanese society, the extent of need, and the extent to which support for assisted reproductive technology is required. We selected this design as a practical and feasible exploratory study.

We added the discussion regarding limitation of this study in "Results and Discussion" section (line 295 to 302).

Statistical plan:

Include more detail regarding missing data handling, adjustment for confounders, and justification for the sample size (even if feasibility-based).

Response: Missing data will be handled using the last observation carried forward method for patient-reported outcomes, and information will continue to be collected as much as possible for survival endpoints and pregnancy-related information throughout the study. Because this is an exploratory study, a formal power calculation will not be performed. To ensure feasibility, the target sample size was set at 10 (a uterine preservation rate of 70% or more is considered acceptable). If the number of events permits, multivariate exploratory analysis will be performed using Cox regression.

Detailed description has been added to 7. Statistical analysis in "Patients and Methods" section (line 209 to 215).

Novelty and contribution:

The use of dose-dense paclitaxel/carboplatin (dd-TC) NAC prior to fertility-sparing surgery could represent a distinctive contribution. Please emphasize this and clearly state how this approach differs from previous reports.

Response: The authors believe that the lack of success of previous NAC studies is due to the low intensity of chemotherapy, the goal of improving prognosis, and the inclusion of cases in which tumors had spread outside the cervix. Regarding chemotherapy regimens in particular, most previous NAC studies have used tri-weekly regimens, whereas this study employed a more intensive regimen called dode-dense paclitaxel. In fact, in the SGSG016 trial, ddTC therapy achieved a response rate of 92%, including a 36% complete response rate, far higher than conventional tri-weekly regimens. We believe this approach is possible only because of the availability of ddTC, a regimen that can effectively shrink tumors.

The response rate and pathological CR rate have been added to the "Results and Discussion" section (line 257 to 258)to further emphasize the usefulness of ddTC therapy.

Clinical implications:

The Discussion should better outline how this feasibility study can inform future multicenter trials and guideline development.

Response: We believe that this study will demonstrate that tumor shrinkage through chemotherapy will enable the introduction of less invasive fertility-sparing surgery and further expand its application. This study will clarify its oncological safety and perinatal prognosis, as well as the type of assisted reproductive technology that is required. Furthermore, we believe that this study will clarify whether this approach is socially acceptable, and, if so, the extent of societal demand for it.

We have added some discussion into "Results and Discussion" section (line 305 to 309 and line 312 to 313).

Minor Comments:

Clarify imaging modalities for determining lymph node status (PET-CT vs. CT).

Response: Before chemotherapy, FDG-PET is used, and before surgery, contrast-enhanced CT (or plain CT if there is an allergy) is used to confirm the presence or absence of lymph node metastasis. Lymph node metastasis is suspected when the SUV max on FDG-PET is greater than 2.5, or when the short axis on CT is greater than 10 mm. The decision will be made separately by one imaging specialist and one gynecological oncologist, and if there is no agreement, the final decision will be made by discussion (line 113 to 119, line 143 to 145, and 167 to 172).

Specify how tumor size is measured and reviewed (MRI criteria, double reading).

Response: Similarly, tumor size will be measured by pelvic contrast MRI (or plain pelvic MRI if the patient has an allergy) four times in total before chemotherapy and after each cycle. The tumor size will be measured separately by one imaging specialist and one gynecological oncologist, and if there is a discrepancy, a decision will be made through discussion (line 113 to 119, line 143 to 145, and 167 to 172).

Correct minor grammatical inconsistencies.

Response: Fixed some grammatical errors and inappropriate wording.

"a normal" to "an intact" (line 142)

"next" to "subsequent" (line 154)

"following" to "upon" (line 218)

"end" to "conclude" (line 219)

"the" to "an" (line 326)

Add line numbering for easier reference during revision.

Response: We apologize for having omitted the line numbers in the previous version. They have now been added.

---

## [Decision Letter · Decision Letter 1]

12 Dec 2025

Dear Dr. Nagao,

Thank you for submitting your manuscript to PLOS ONE. After careful consideration, we feel that it has merit but does not fully meet PLOS ONE’s publication criteria as it currently stands. Therefore, we invite you to submit a revised version of the manuscript that addresses the points raised during the review process.

We look forward to receiving your revised manuscript.

Kind regards,

Kazunori Nagasaka

Academic Editor

PLOS One

Journal Requirements:

Additional Editor Comments:

Dear Prof. Nagao,

The manuscript has improved, but several issues require revision.

As mentioned by Reviewer 1, the description of SLNB availability in Japan is outdated; RI-based SLN mapping has been reimbursed since 2023, and ICG-guided SLNB will be covered in 2025.

This should be corrected and discussed in relation to the choice of full lymphadenectomy. Redundancy between sections remains, and the Discussion should focus more clearly on risks and decision criteria when combining NAC with fertility-sparing surgery. The limitations should also note the lack of pathological nodal staging before NAC.

We look forward to receiving your revised manuscript soon.

Sincerely,

Kazunori Nagasaka

Reviewers' comments:

Reviewer's Responses to Questions

**Comments to the Author**

1. Does the manuscript provide a valid rationale for the proposed study, with clearly identified and justified research questions?

Reviewer #1: Yes

Reviewer #2: Yes

Reviewer #3: Yes

Reviewer #5: Yes

2. Is the protocol technically sound and planned in a manner that will lead to a meaningful outcome and allow testing the stated hypotheses?

Reviewer #1: Yes

Reviewer #2: Yes

Reviewer #3: Yes

Reviewer #5: Yes

3. Is the methodology feasible and described in sufficient detail to allow the work to be replicable?

Reviewer #1: Yes

Reviewer #2: Yes

Reviewer #3: Yes

Reviewer #5: Yes

4. Have the authors described where all data underlying the findings will be made available when the study is complete?

Reviewer #1: Yes

Reviewer #2: No

Reviewer #3: Yes

Reviewer #5: Yes

5. Is the manuscript presented in an intelligible fashion and written in standard English?

Reviewer #1: Yes

Reviewer #2: Yes

Reviewer #3: Yes

Reviewer #5: Yes

You may also provide optional suggestions and comments to authors that they might find helpful in planning their study.

Reviewer #1: The authors have made substantial improvements in their revised manuscript. The integration of recent GOG trial data, clarification of methodological details, and enhanced justification for fertility-sparing treatment in bulky IB2–IB3 disease are appreciated. However, several important issues remain before the manuscript is suitable for publication.

1. Sentinel Lymph Node Biopsy (SLNB): inaccurate and outdated statements

The revised manuscript states that sentinel lymph node biopsy is not widely available or insured in Japan.

This is no longer accurate. Current status in Japan (important for this protocol)

Since 2023, SLN mapping using radioisotope tracers (RI) has been covered by national insurance.

From summer 2025, indocyanine green (ICG)-guided SLN mapping will also be reimbursed, allowing SLN detection without radiation exposure.

As a result, SLNB using ICG fluorescence is expected to become widely feasible, especially in young and fertility-oriented patients.

Given these developments, the authors’ statements underestimate the current and near-future accessibility of SLNB in Japan.

Required revisions

Correct the description of insurance status for SLNB (RI and ICG).

Update the Discussion to reflect how the availability of radiation-free ICG mapping may influence future iterations of the protocol.

Comment briefly on why full pelvic lymphadenectomy was selected despite these recent changes, and how SLNB + ultrastaging could reduce morbidity in fertility-sparing candidates.

This correction is essential for methodological accuracy and for aligning the protocol with evolving clinical practice.

2. Redundancy between Introduction and Discussion

Several sections still repeat the same information:

(a) Tumors ≤2 cm and evidence from SHAPE / ConCerv / GOG-278

These data are described extensively in both Introduction and Discussion. In the Discussion, they should appear only to support interpretation of the protocol’s design—not as repeated background.

(b) Rationale for dd-TC chemotherapy

Response rates and pCR values appear in both sections almost unchanged.

Recommendation

Streamline the Discussion by removing repeated background and focusing on how these external results inform safety expectations and feasibility for IB2–IB3 tumors.

3. Dual Experimental Approach: more explicit interpretation needed

The authors now acknowledge that the protocol combines two investigational elements:

(1) NAC for tumor downstaging, and

(2) fertility-sparing surgery.

However, the Discussion still leans heavily on general evidence supporting each modality rather than explaining:

Which specific risks remain unknown when both interventions are combined

How response criteria, conization pathology, and radiologic reassessment mitigate those risks

What constitutes a safety threshold to abandon fertility preservation and proceed to standard treatment

Explicitly stating these points will improve transparency and strengthen the scientific rationale.

4. Limitations section: incomplete

Although sample size, single-center design, and short follow-up are discussed, one limitation requires clearer acknowledgment:

Lack of pathological nodal staging prior to NAC may create staging uncertainty, because chemotherapy can suppress but not eradicate nodal metastases.

Given the increasing availability of SLNB (RI and ICG), the omission should be explicitly recognized as a limitation.

5. Length and focus of the Discussion

The Discussion remains relatively long and contains extensive summaries of prior trials.

A more effective structure would emphasize:

What new insights the feasibility study aims to generate

How the design specifically addresses the safety of fertility preservation in bulky IB2–IB3 tumors

How the results could inform future multicenter studies or guideline development

Reducing narrative background and highlighting the protocol’s unique contribution will enhance readability.

6. Strengths of the revision

Several improvements should be acknowledged:

Clearer surgical criteria and pathological evaluation

Strengthened description of adverse event monitoring

Integration of recent GOG trial literature

Better explanation of the intended contribution to fertility-sparing oncologic management

These revisions have significantly improved the manuscript.

Overall Recommendation

The manuscript has improved, but additional revision is still required.

In particular:

Update the description of SLNB insurance coverage in Japan, including the 2025 adoption of ICG fluorescence mapping, which eliminates radiation exposure.

Reduce redundancy between sections.

Clarify limitations and sharpen the rationale for the dual experimental design.

Addressing these points will greatly enhance the accuracy, clarity, and scientific rigor of the protocol.

Reviewer #2: All comments have been adequately addressed.

Reviewer #3: The questions were responded and the improvements were done.

Aim is clear

Study design description is accurate

Eligibility criteria clear

ddTC regimen description acceptable

Primary and Secondary endpoints are appropriate

AE reporting (CTCAE v5.0) is standard

Reviewer #5: I would like to thank the authors for answering the various issues. They have response to all of issue.

**Do you want your identity to be public for this peer review?** For information about this choice, including consent withdrawal, please see our Privacy Policy

Reviewer #1: No

Reviewer #2: No

Reviewer #3: No

Reviewer #5: No

---

## [Author Response · Author response to Decision Letter 2]

16 Dec 2025

December 16, 2025

Kazunori Nagasaka

Academic Editor

PLOS ONE

Subject: Resubmission of the revised manuscript (PONE-D-25-48071)

Dear Dr. Nagasaka,

We sincerely thank you for the constructive and insightful comments, which were extremely helpful in further improving the accuracy, clarity, and clinical relevance of our manuscript “Fertility-sparing surgery with neoadjuvant chemotherapy in early and locally advanced cervical cancer: A clinical protocol.”.

We have carefully revised the manuscript in accordance with the Editor’s suggestions. In particular, we have updated the description of the current insurance coverage and clinical availability of sentinel lymph node biopsy (SLNB) in Japan, reduced redundancy between sections, clarified the decision-making criteria and risks associated with combining neoadjuvant chemotherapy with fertility-sparing surgery, and explicitly addressed the lack of pathological nodal staging prior to neoadjuvant chemotherapy as a study limitation.

Response to Reviewers

Comment 1: SLNB: inaccurate and outdated statements

Response: We thank the reviewer for this important and timely comment. At the start of this study (IRB review), only Techne® Phytate used in the RI method was approved in Japan, so sentinel lymph node biopsy was not included in the study. As you pointed out, indocyanine green (ICG), a fluorescent dye tracer, was already approved in 2025, and we have heard that the lymph node biopsy procedure is expected to be covered by insurance in 2026. Actually, our facility introduced sentinel lymph node biopsy into clinical practice in December 2025. The revised manuscript now states the latest information on SNL mapping (Line 263 to 265).

Comment 2: Redundancy between Introduction and Discussion

Response: We appreciate this helpful suggestion. We have substantially streamlined the Discussion by removing repeated background information regarding tumors ≤2 cm and dd-TC chemotherapy. The Discussion now focuses on interpretation of the protocol design, safety considerations, and applicability to IB2–IB3 disease, while detailed background evidence is retained in the Introduction (Line 58 to 64 and 81 to 84).

Comment 3: Dual experimental approach

Response: We agree that explicit interpretation is essential. The Discussion has been revised to clearly describe the specific risks associated with combining neoadjuvant chemotherapy and fertility-sparing surgery, the predefined radiologic and pathological safety criteria, and the thresholds at which fertility preservation is abandoned in favor of standard treatment (Line 252 to 259).

Comment 4: Limitations section

Response: We have added a specific limitation acknowledging the lack of pathological nodal staging prior to neoadjuvant chemotherapy and discussed the potential for residual occult nodal disease despite radiologic response. This point is now clearly stated in the Discussion section (Line 265 to 270).

Comment 5: Length and focus of the Discussion

Response: The Discussion has been reorganized to emphasize the unique contribution of this feasibility study, its safety framework, and its potential implications for future multicenter trials and guideline development. Narrative background has been reduced accordingly in Discussion section.

We sincerely thank the reviewer for recognizing the strengths of the revised manuscript and for providing constructive guidance that has significantly improved its scientific rigor and clarity. We believe that the revised version now fully addresses all editorial and reviewer comments, and has been significantly improved in clarity, structure, and scientific rigor. We sincerely hope that this revised manuscript will be suitable for publication in PLOS ONE.

Thank you very much for your kind consideration.

Sincerely,

Shoji Nagao, MD, PhD

Department of Obstetrics and Gynecology

Okayama University, Japan

E-mail: s_nagao@okayama-u.ac.jp

---

## [Decision Letter · Decision Letter 2]

18 Dec 2025

Dear Dr. Nagao,

Thank you for submitting your manuscript to PLOS ONE. After careful consideration, we feel that it has merit but does not fully meet PLOS ONE’s publication criteria as it currently stands. Therefore, we invite you to submit a revised version of the manuscript that addresses the points raised during the review process.

We look forward to receiving your revised manuscript.

Kind regards,

Kazunori Nagasaka

Academic Editor

PLOS One

Journal Requirements:

Additional Editor Comments:

Dear Dr.Nagao,

Thank you for submititng your manuscript to Plos One.

A reviewer has commented to your revised manuscript.

Please revise the content accordingly, and we look forward to receiving your manuscript soon.

Sincerely,

Kazunori Nagasaka

Reviewers' comments:

Reviewer's Responses to Questions

**Comments to the Author**

1. Does the manuscript provide a valid rationale for the proposed study, with clearly identified and justified research questions?

Reviewer #1: Yes

2. Is the protocol technically sound and planned in a manner that will lead to a meaningful outcome and allow testing the stated hypotheses?

Reviewer #1: Yes

3. Is the methodology feasible and described in sufficient detail to allow the work to be replicable?

Reviewer #1: Yes

4. Have the authors described where all data underlying the findings will be made available when the study is complete?

Reviewer #1: Yes

5. Is the manuscript presented in an intelligible fashion and written in standard English?

Reviewer #1: Yes

You may also provide optional suggestions and comments to authors that they might find helpful in planning their study.

Reviewer #1: The authors have carefully and thoroughly revised the manuscript in response to the previous reviewer comments. The revised version shows clear improvement in structure, clarity, and methodological transparency. In particular, the updated description of sentinel lymph node biopsy in the Japanese clinical context, the streamlined Discussion, and the explicit definition of safety criteria and interim analysis substantially strengthen the protocol.

Overall, this is a well-designed and clearly presented Phase II feasibility study addressing an important and clinically relevant unmet need in young women with cervical cancer who desire fertility preservation. The manuscript is suitable for publication after minor revisions aimed at further clarification rather than substantive modification.

1. Interpretation of the primary endpoint

The use of uterine preservation as the primary endpoint is appropriate for a feasibility study. However, a brief clarification of how this endpoint should be interpreted alongside oncologic outcomes (e.g., short-term RFS) would improve clinical interpretability.

Suggested action: Add one or two sentences in the Discussion clarifying that uterine preservation is evaluated in conjunction with oncologic safety endpoints, rather than as an isolated success measure.

2. Imaging-based assessment after neoadjuvant chemotherapy

The manuscript appropriately explains the rationale for relying on post-NAC imaging and subsequent pelvic lymphadenectomy. For balance, it would be helpful to briefly acknowledge the inherent limitations of imaging-based response assessment, including the possibility of occult residual disease.

Suggested action: Add a short statement in the Discussion noting this limitation.

3. Sample size justification

The feasibility nature of this study and the small sample size are acceptable. A short explanation of the practical or ethical considerations that informed the choice of a target sample size of 10 patients would enhance transparency.

Suggested action: Provide a brief clarification in the Sample Size or Discussion section.

4. Ethical transparency

Given the dual experimental nature of the proposed strategy, the ethical framework is generally well described. A brief statement emphasizing that patients are explicitly informed of the non-standard and exploratory aspects of this approach would further strengthen the protocol.

Suggested action: Add a single sentence to the Ethical Considerations section.

**Do you want your identity to be public for this peer review?** For information about this choice, including consent withdrawal, please see our Privacy Policy

Reviewer #1: **Yes: ** Tsukasa Baba

---

## [Author Response · Author response to Decision Letter 3]

22 Dec 2025

December 19, 2025

Kazunori Nagasaka

Academic Editor

PLOS ONE

Subject: Resubmission of the revised manuscript (PONE-D-25-48071)

Dear Dr. Nagasaka,

Thank you very much for the opportunity to revise our manuscript. We are also grateful to the reviewer, Dr. Baba, for his constructive and encouraging comments on our study. We have carefully addressed all the minor revisions suggested by the reviewer.

We hope that these revisions further enhance the clarity and transparency of our protocol and that the manuscript is now suitable for publication in PLOS ONE.

Response to Reviewers

Comment 1: Interpretation of the primary endpoint

The use of uterine preservation as the primary endpoint is appropriate for a feasibility study. However, a brief clarification of how this endpoint should be interpreted alongside oncologic outcomes (e.g., short-term RFS) would improve clinical interpretability. Suggested action: Add one or two sentences in the Discussion clarifying that uterine preservation is evaluated in conjunction with oncologic safety endpoints, rather than as an isolated success measure.

Response: We agree with the reviewer’s suggestion. While the primary goal is feasibility as measured by the uterine preservation rate, this is only clinically meaningful when achieved without compromising oncologic safety. We have added a sentence to the Discussion to clarify this integrated interpretation. We have added the sentence " Importantly, successful uterine preservation in this study is not viewed as an isolated metric but is evaluated in close conjunction with oncological safety endpoints, such as 2-year RFS, to ensure that fertility-sparing does not jeopardize patient survival." to the Results and Discussion section (Line 264 to 267).

Comment 2: Imaging-based assessment after neoadjuvant chemotherapy

The manuscript appropriately explains the rationale for relying on post-NAC imaging and subsequent pelvic lymphadenectomy. For balance, it would be helpful to briefly acknowledge the inherent limitations of imaging-based response assessment, including the possibility of occult residual disease. Suggested action: Add a short statement in the Discussion noting this limitation.

Response: We acknowledge the inherent limitations of imaging-based response assessment as pointed out by the reviewer. We have added a statement in the Discussion noting the possibility of occult residual disease that may not be captured by imaging alone. WE have added the sentence " However, we recognize the inherent limitations of imaging-based assessment, as PET-CT and MRI may not fully detect occult micro-residual disease or small-volume nodal involvement after NAC." to the Results and Discussion section (Line 271 to 273).

Comment 3: Sample size justification

The feasibility nature of this study and the small sample size are acceptable. A short explanation of the practical or ethical considerations that informed the choice of a target sample size of 10 patients would enhance transparency. Suggested action: Provide a brief clarification in the Sample Size or Discussion section.

Response: Thank you for this suggestion. The target sample size of 10 was determined based on the practical challenges of recruiting eligible patients at a single institution and the ethical imperative to limit exposure to this novel dual-experimental strategy until preliminary safety is confirmed. We have clarified this in the Sample Size section. We have replaced the sentence " To ensure feasibility, the target sample size was set at 10 participants, assuming that a uterine preservation rate of ≥70% would be considered clinically acceptable." to " The target sample size of 10 was chosen based on practical considerations regarding the rarity of FIGO stage IB2–IB3 patients seeking fertility preservation at a single center, as well as the ethical necessity of limiting participant exposure to this novel approach until the planned interim safety analysis is completed." at the Sample Size section (Line 220 to 223).

Comment 4: Ethical transparency

Given the dual experimental nature of the proposed strategy, the ethical framework is generally well described. A brief statement emphasizing that patients are explicitly informed of the non-standard and exploratory aspects of this approach would further strengthen the protocol. Suggested action: Add a single sentence to the Ethical Considerations section.

Response: As suggested, we have added a sentence to the Ethical Considerations section to emphasize that all participants are explicitly informed of the non-standard and exploratory nature of the proposed strategy during the informed consent process. We have added the sentence " During the informed consent process, it is explicitly emphasized to all participants that this treatment strategy is non-standard and exploratory in nature, involving potential risks associated with the dual-experimental approach." to the Ethical Considerations section (Line 236 to 239).

Sincerely,

Shoji Nagao, MD, PhD

Department of Obstetrics and Gynecology

Okayama University, Japan

E-mail: s_nagao@okayama-u.ac.jp

---

## [Decision Letter · Decision Letter 3]

30 Dec 2025

Fertility-sparing surgery with neoadjuvant chemotherapy in early and locally advanced cervical cancer: A clinical protocol

PONE-D-25-48071R3

Dear Dr. Nagao,

We’re pleased to inform you that your manuscript has been judged scientifically suitable for publication and will be formally accepted for publication once it meets all outstanding technical requirements.

Kind regards,

Kazunori Nagasaka

Academic Editor

PLOS One

Additional Editor Comments (optional):

Dear Prof. Nagao,

Congratulations.

I am pleased to inform you that your manuscript has been accepted for publication in PLOS ONE.

The revisions have clearly strengthened the clarity, methodological transparency, and ethical rigor of the study.

I wish you continued success in your important research endeavors.

Warmest congratulations to you and your team.

Sincerely,

PLOS ONE

Kazunori Nagasaka

Reviewers' comments:

Reviewer's Responses to Questions

**Comments to the Author**

1. Does the manuscript provide a valid rationale for the proposed study, with clearly identified and justified research questions?

Reviewer #1: Yes

Reviewer #2: Yes

2. Is the protocol technically sound and planned in a manner that will lead to a meaningful outcome and allow testing the stated hypotheses?

Reviewer #1: Yes

Reviewer #2: Yes

3. Is the methodology feasible and described in sufficient detail to allow the work to be replicable?

Reviewer #1: Yes

Reviewer #2: Yes

4. Have the authors described where all data underlying the findings will be made available when the study is complete?

Reviewer #1: Yes

Reviewer #2: No

5. Is the manuscript presented in an intelligible fashion and written in standard English?

Reviewer #1: Yes

Reviewer #2: Yes

You may also provide optional suggestions and comments to authors that they might find helpful in planning their study.

Reviewer #1: The authors have responded to all reviewer comments in a thorough, thoughtful, and transparent manner. Each point has been appropriately addressed with concise revisions that improve the clarity, interpretability, and ethical transparency of the manuscript without altering its feasibility-focused intent.

Specifically, the added clarification regarding the interpretation of uterine preservation in conjunction with oncologic safety endpoints strengthens the clinical relevance of the primary endpoint. The acknowledgment of the limitations of imaging-based assessment after neoadjuvant chemotherapy provides appropriate balance, and the revised explanation of the sample size justification enhances methodological transparency. In addition, the explicit statement regarding informed consent and the exploratory nature of the strategy further reinforces the ethical rigor of the study.

Overall, these revisions meaningfully improve the manuscript. I have no further substantive comments and recommend acceptance of the manuscript in its current form.

Reviewer #2: All comments have been adequately addressed.

**Do you want your identity to be public for this peer review?** For information about this choice, including consent withdrawal, please see our Privacy Policy

Reviewer #1: **Yes: ** Tsukasa Baba

Reviewer #2: No

---

## [Editor Report · Acceptance letter]

PONE-D-25-48071R3

PLOS One

Dear Dr. Nagao,

I'm pleased to inform you that your manuscript has been deemed suitable for publication in PLOS One. Congratulations! Your manuscript is now being handed over to our production team.

Kind regards,

on behalf of

Professor Kazunori Nagasaka

Academic Editor

PLOS One